# Intakes of Lean Proteins and Processed Meats and Differences in Mental Health between Rural and Metro Adults 50 Years and Older

**DOI:** 10.3390/nu16183056

**Published:** 2024-09-11

**Authors:** Nathaniel R. Johnson, Sherri N. Stastny, Julie Garden-Robinson

**Affiliations:** 1Department of Nutrition and Dietetics, University of North Dakota, Northern Plains Center for Behavioral Research, Room 340E, 430 Oxford Street, Grand Forks, ND 58202, USA; 2Department of Health, Nutrition, and Exercise Sciences, North Dakota State University, E. Morrow Lebedeff Hall, 1310 Centennial Boulevard, Fargo, ND 58102, USA; sherri.stastny@ndsu.edu; 3Food and Nutrition Extension, North Dakota State University, Katherine Kilbourne Burgum Family Life, 4-H Center, 1400 Centennial Boulevard, Fargo, ND 58102, USA; julie.gardenrobinson@ndsu.edu

**Keywords:** older adults, dietary protein, depression, anxiety, processed meat

## Abstract

Mental health disparities exist between rural and metro areas of the United States. Differences in dietary intake may contribute to these disparities. We examined differences in dietary intake and mental health between those 50 years and older (*n* = 637) living in rural counties to those living in metro counties in North Dakota and the relationship between dietary intake to days with depression or anxiety. A survey was conducted throughout North Dakota. Items were modified from other surveys, such as the Behavioral Risk Factor Surveillance System questionnaires and the National Health Interview Survey Cancer Control Supplement Dietary Screener Questionnaire. Comparing medians, individuals more likely to be unable to perform normal daily activities due to mental health (*p* = 0.009) resided in rural areas instead of metro areas. Those living rurally also ate more processed meats (*p* = 0.005), while trending toward less lean protein intake (*p* = 0.056). Multinomial regression analyses controlling for covariates revealed that lean protein intake and fruit intake were inversely associated with days with depression and anxiety (all *p* < 0.05), whereas processed meat intake was positively associated with anxiety (*p* = 0.005). Clinicians working with older adults residing in rural areas should emphasize substituting lean proteins for processed meats.

## 1. Introduction

Although perhaps best known for its relationship with health issues such as cardiovascular disease [1,2,3], obesity [4], and type 2 diabetes mellitus [5,6], dietary intake is also a potent driver of mental health. For instance, those in the highest tertile of branched chain amino acid intake, a marker of dietary protein quality [7], had 34.0% and 24.0% decreased risk of anxiety and depression, respectively [8]. It has been reported that dietary fiber intake, another indicator of dietary quality, although in this case related to fruit, vegetables, and whole grains, was related to decreased odds of depression; a 5 g per day increase in dietary fiber corresponded to a 5% decrease in odds of depression [9]. Thus, dietary intake is an important component of not just physical health, but mental health too.

Unfortunately, aging negatively impacts both dietary intake [10] and mental health [11]. The consumption of nutrient-dense foods is hindered by problems that occur with aging, such as poor dentition, living alone, limited resources and access to healthy food, decreased vision, taste, and olfactory sense, reduced appetite, and other physical or environmental constraints [10]. In addition, older adults often eat less than they did when they were younger to counterbalance decreases in physical activity and resting metabolic rate that can be part of aging [12]. Due to decreased intake, 50.0% of older adults in the United States meet dietary protein intake recommendations and less than 17.0% meet dietary fiber recommendations [13]. As previously noted, dietary intakes of protein [8] and fiber [9] were positively related to improved mental health. Beyond the effects of not meeting dietary protein recommendations on mental health [8], processed meats (i.e., cured, salted, smoked, or otherwise further processed fresh meat) are a main source of protein for older adults living in the United States [14]. Processed meats are typically high in both saturated fat and sodium and other added ingredients, while providing less protein than lean protein sources (e.g., lean beef, pork, poultry, dairy products and others not processed), given the same serving size [15]. Evidence suggests that saturated fat intake worsens hippocampal function, potentially leading to mental health problems [16]; in fact, the intake of some saturated fats has been related to increased prevalence of depression [17].

North Dakota (ND) is one of the most rural, sparsely populated states in the United States. According to the 2020 United States census report, there were 779,094 individuals in residency, with 16.7% age 65 or older, living across a total land area of a little more than 110,000 km^2^ [18]. This equates to a population density of approximately seven people per square kilometer. While more populated metro areas in the United States have grocery retail outlets, the average small town or rural community may be limited to only convenience stores (i.e., gas stations with retail food) [19] where choices are limited to processed foods, like processed meats [20]. In support of this, as of 2019, 45.6% of those living in metro counties in ND had low access to healthful foods compared to 56.8% or those living in non-metro or rural counties [21]. This difference in the food environments between rural and metro areas may lead to disparities in health outcomes; for example, North Dakotans living in rural counties are more likely to have diabetes (9.6% vs. 8.5%) and have a lower life expectancy at birth than those living in metro counties (79.3 years vs. 80.6 years) [21].

The *Nourish* program is a face-to-face and online United States Department of Agriculture-funded Food and Nutrition Extension (Ext) education program for middle-aged to older North Dakotans emphasizing healthy aging [22]. The program covers many topics, including stress, sleep, and exercise, but its focus is on healthful dietary intake. The *Nourish* program was launched in 2013 and was recently revamped. As part of this update process, North Dakotans older than 50 years of age were surveyed about their health and dietary intake. We sought to examine differences in self-reported mental health (i.e., days with depression and anxiety) and dietary intake between North Dakotans living in rural counties versus those living in metro counties.

## 2. Materials and Methods

### 2.1. Study Protocol

To evaluate the ongoing needs of the *Nourish* program’s updates, a 79-item survey was administered and delivered throughout ND during spring, 2023. Only those who reported being 50 years or older were eligible to complete the survey. Most items for the survey were modified with approval from other surveys, such as Behavioral Risk Factor Surveillance System questionnaires [23] and the National Health Interview Survey Cancer Control Supplement Dietary Screener Questionnaire [24]. The survey was disseminated in e-newsletters, through state news releases, through county-based listservs, through social media (e.g., Facebook), and through a nutrition column that appears in 50 newspapers and online (Qualtrics, Provo, UT, USA). It also was made available as a paper copy, and 500 copies were distributed to county extension offices throughout the state to gather responses from in-person events. Participants were offered a chance at winning small prizes for participating. Only those who answered 90% or more of the survey questions were considered valid responses. Coded values used for statistical analyses are described below parenthetically. The study was approved by North Dakota State University’s Institutional Review Board (IRB) #IRB0004482. Informed consent information was provided before completion of the survey. This program was supported by the USDA National Institute of Food and Agriculture as a Rural Health and Safety Education project.

Although the survey was technically available nationally, the focus of the *Nourish* program is its home state, ND, where the survey was intended to be delivered to inform future program updates and developments. As a result, data regarding the county in which a participant resides is limited to ND. This is important, as living rurally was defined using the county in which participants reported residency; more specifically, rural counties were defined using the National Center for Health Statistics’ Urban–Rural Classification Scheme for Counties [25]; that is, only non-metro counties were considered rural counties (0 = rural county; 1 = metro county). According to the most recent classification by the National Center for Health Statistics [25], there are only six metro counties in North Dakota: Burleigh, Cass, Grand Forks, Morton, Oliver, and Sioux counties. Those who were not ND residents were excluded from analyses, as it was not possible to classify these respondents as living in a rural or metro county.

Demographic questions included self-reported age, sex, race and ethnicity, cohabitation status, education level, and household income. Age (1 = 50–55 years, 2 = 56–61 years, 3 = 62–64 years, 4 = 65–70 years, and 5 = 71+ years), education level (1 = “Did not finish high school”, 2 = “High school graduate”, 3 = “Some college” 4 = “Associate Degree”, 5 = “College Graduate (B.S. or B.A. degree)”, 6 = “Master’s degree”, and 7 = “Completed post-graduate (M.D., Ph.D.)”), and household income (1 = “$0–24,999”, 2 = “$25,000–49,999”, 3 = “$50,000–74,999” 4 = “$75,000–99,999”, 5 = “$100,000–149,999”, and 6 = “≥$150,000) were measured as ordinal variables. Sex, cohabitation status, and race and ethnicity were treated as categorical variables.

Mental health status was assessed using items derived from versions of the Behavioral Risk Factor Surveillance System questionnaire [23]: (1) “Regarding your MENTAL health (stress, depression, emotions, etc.) in the past 30 days, how many days would you consider your mental health ‘not good?’”, (2) “Regarding your MENTAL health, in the past 30 days, how many days were you unable to perform your usual, daily activities (self-care, work, recreation, etc.)?”, (3) “In the past 30 days, for how many days have you felt SAD, BLUE, or DEPRESSED?”, and (4) “In the past 30 days, for how many days have you felt WORRIED, TENSE, or ANXIOUS?” These items were answered using a six-point ordinal scale (1 = 0–5 days, 2 = 6–10 days, 3 = 11–15 days, 4 = 16–20 days, 5 = 21–25 days, and 6 = 26–30 days).

Dietary intake was estimated using a study specific food frequency questionnaire with most items modified from the National Health Interview Survey Cancer Control Supplement Dietary Screener Questionnaire [24]. In addition to assessing the intake of soy and other beans, processed meats, fruits, vegetables, and leafy greens which were modified from this questionnaire [24], our questionnaire included two more items about sources of dietary protein. One item asks about lean protein intake, “How often do you eat a meal or snack that includes a source of lean protein (lean meat, poultry, fish and other seafood, eggs, dairy, soy and other beans, and other substitutes)?”, whereas the other asks about nut and seed intake, “How often do you eat a meal or snack that includes nuts and seeds?” Items assessing dietary intake were answered using a seven-point ordinal scale (1 = “Never”, 2 = “Rarely”, 3 = “Less than once per week”, 4 = “About once per week”, 5 = “Every two or three days”, 6 = “Once per day”, and 7 = “More than once per day”).

### 2.2. Statistical Analyses

Statistical analyses were performed using SPSS Version 27 (IBM Corp, Armonk, NY, USA). Each statistical analysis contained the maximum number of participants, but the number of participants varied across analyses, as those missing one or more values were excluded from that analysis.Descriptive statistics were presented as frequencies. Normal distribution for variables was not assumed, so nonparametric tests were used to examine differences between those living in rural, non-metro counties and those living in metro counties. Mood’s test of independent samples’ medians was used for ordinal variables and the Mann–Whitney U test was used for categorical variables. For the categorical variable sex, those who identified as “other” were not included (0 = female; 1 = male) as only six participants identified as sex other than male or female. For cohabitation status, those who live with others were treated as one category, including both those who live with others and those who live with spouses, whereas those living alone (0 = live with others; 1 = lives alone) were treated as a separate category. Lastly, due to fact that more than 80% of the sample identified as non-Hispanic white, for race and ethnicity non-Hispanic white was considered to be one category while all other races and ethnicities were treated as a second category (0 = all other races and ethnicities; 1 = non-Hispanic white). These codes for categorical variables were also used in subsequent analyses.

Mixed linear models were used to examine the association between self-reported days with depression and anxiety with self-reported “not good” mental health days and days where participants reported being unable to perform normal activities due to poor mental health, while controlling for non-dietary covariates, namely age, sex, race and ethnicity, cohabitation status, education level, and household income. An interaction between depression and anxiety was hypothesized, such that those with greater amounts of depression and anxiety would be more likely to report a greater number of “not good” mental health days and a greater number of days where they were unable to perform activities due to poor mental health.

Simple and multinomial regression models examined the impact of dietary intake on self-reported days with depression and anxiety. First, single dietary variables were associated with either marker of mental health using simple regression models. Then, multinomial models including non-dietary covariates were used. Non-dietary covariates were age, sex, race and ethnicity, cohabitation status, education level, and household income, and these variables were entered into regression models with individual dietary intake variables. Finally, complete multinomial models where the aforementioned covariates and all dietary intake variables were entered together were used to investigate the roles of dietary intake variables while controlling for other aspects of dietary intake. The graphical abstract was produced using Spearman’s Rho correlation. The alpha value was set at *p* = 0.05, and two-tailed tests were used for all analyses.

## 3. Results

### 3.1. Descriptive Statistics

There were 2136 responses to the survey. However, it was determined that 471 responses were incomplete (i.e., response rate <90%) and were excluded. Only 637 adults 50 years and older from ND completed the survey. Most, 504 (79.1%), were from rural counties, whereas 133 (20.9%) were from metro counties. Self-reported age, race and ethnicity, education, income, and cohabitation status are displayed in Table 1, partitioned into rural and metro counties. Across the whole sample, the majority, 79.1%, were between 50 and 65 years of age; 54.5% were female; 81.3% identified as non-Hispanic white; 61.2% held an associate’s degree or higher; 54% reported incomes of USD 75,000 or more; and lastly, 22.0% lived alone.

As part of the questionnaire, participants were probed with four different questions to assess self-rated mental health. Responses to these questions are described in Table 2, again by rural and metro county residency. More than half of the entire sample reported the fewest number of days possible (i.e., zero to five days) for all four items assessing mental health status in the last 30 days. More specifically, a total of 370 participants, or 58.1%, indicated zero to five days of “not good” mental health, 452 (71.0%) reported zero to five days of being unable to do activities due to mental health, 369 (57.9%) had zero to five days with depression, and 323 (50.7%) had zero to five days with anxiety.

Seven questions were used to assess self-reported dietary intake. Responses are detailed in Table 3 by rural and metro county residency. A majority of the whole sample reported eating lean proteins (57.9%), fruits (60.9%), and vegetables (67.3%) at least once a day, whereas fewer answered that they consumed leafy greens (47.1%), nuts and seeds (31.7%), processed meats (28.9%), and beans (17.7%) at least once a day.

### 3.2. Comparisons between Those Residing in Rural versus Metro Counties

Table 4 describes comparisons between those living in rural counties compared to those living in metro counties. Those living in rural counties were younger (*p* = 0.008), less likely to identify as non-Hispanic white (*p* < 0.001), and had lower levels of educational attainment (*p* < 0.001). Significant differences were not found in self-reported sex, cohabitation status, or household income. Those living in rural counties reported a greater number of days where they were unable to perform activities due to mental health (*p* = 0.009), even though they reported an equivalent number of “not good” mental health days compared to those living in metro counties. Intake of processed meats (*p* = 0.005) and beans (*p* = 0.017) were significantly greater in rural respondents. In addition, rural participants reported close to statistically significantly lower intake of lean proteins (*p* = 0.056).

### 3.3. The Interaction between Days with Depression and Days with Anxiety on Mental Health Status

Mixed models showed that there is an interaction between anxiety and depression with reported “not good” mental health days (Model: R^2^ = 0.501, F_29,559_ = 19.365, *p* < 0.001; interaction: F_13,559_ = 1.970, *p* = 0.021) and the number of days participants reported being unable to do normal activities due to mental health (Model: R^2^ = 0.328, F_29,559_ = 9.417, *p* < 0.001; interaction: F_13,559_ = 1.849, *p* = 0.033). These interactions are depicted in Figure 1 and Figure 2, respectively. Main effects for anxiety and depression were also found in both mixed models (all *p* < 0.05). Together, this indicates that as days with anxiety and/or depression increase, so does participants’ reporting of “not good” mental health days and days not being able to perform activities due to mental health.

### 3.4. Associations between Dietary Intake with Depression or Anxiety

Associations between aspects of dietary intake with depression and anxiety are described in Table 5 and Table 6. Fully adjusted analyses for non-dietary covariates and dietary intake revealed that lean protein intake and fruit intake are inversely associated with days with depression (β for lean protein = −0.117 ± 0.040, *p* = 0.004; β for fruit = −0.096 ± 0.038, *p* = 0.011) and anxiety (β for lean protein = −0.086 ± 0.040, *p* = 0.031; β for fruit = −0.095 ± 0.038, *p* = 0.012). Bean intake was inversely related to days with anxiety (β = −0.092 ± 0.037, *p* = 0.013), whereas processed meat intake was positively associated with anxiety (β = 0.089 ± 0.040, *p* = 0.005).

## 4. Discussion

Our findings indicate a mental health disparity between those 50 years and older living in rural North Dakota counties compared to those living in metro-counties. Those living in rural counties were more likely to not be able to perform normal daily activities due to mental health than those living in metro counties, despite the two groups reporting similar amounts of “not good” mental health days. This is in line with other research regarding mental health disparities between rural and urban environments which reported similar prevalences of mental health issues between the two settings but worse outcomes for those in rural areas [26]. Our study’s results also indicate that days with depression and anxiety are related to greater reports of the number of days that one is unable to perform normal activities due to mental health. Although various aspects of the rural environment play a role in mental health disparities [26], such as a lack of mental health providers in rural communities [27], our results support the role of differences in dietary intake as environmental factors that lead to disparities in mental health outcomes between rural and metro communities. More specifically, rural older adults consumed more processed meats, while trending toward less lean protein intake. This is important, as lean protein intake was inversely associated with depression and anxiety, whereas processed meat intake was positively associated with anxiety.

Although both processed and whole meats may provide high-quality dietary protein [7] rich in essential amino acids, whose intake is related to decreased odds of depression [8], there are key nutritional differences between lean sources of dietary protein and processed meats. For example, a 100 g serving of Italian style pork salami provides 21.7 g of protein, 13.1 g of saturated fat, and 1890 mg of sodium, whereas an equivalent serving of 100 g of roasted chicken breast without the skin provides 31.0 g protein, 1.0 g of saturated fat, and 74 mg of sodium [28]. Problematically, the intake of some saturated fats [17] and liking the taste of salt [29] are related to poorer mental health. However, fresh meats have the disadvantage of having shelf lives of only three to five days, which pales in comparison to the shelf lives of some processed meats [30]. As those in rural communities are often farther away from grocery stores [19], foods choices are often more limited, leading to less healthful food choices [20]. Interestingly, we did not find significant differences in fruit or vegetable intakes between those living in rural or metro counties, despite seemingly worse food access in non-metro counties [21]. In fact, those in rural counties consumed more beans than those in metro counties.

In addition to the associations between lean protein and processed meat intakes with mental health, we also found that intake of beans and fruits was inversely associated with days with anxiety, with fruit intake also being inversely associated with days with depression. These results are ambiguous compared to findings from a meta-analysis that indicated that dietary fiber intake is related to decreased odds of depression [9]. This meta-analysis found that dietary fiber from vegetables, but not fruit, was related to decreased odds of depression [9], which is in contrast to our findings where fruit intake, but not vegetable intake, was inversely related to days with depression in fully adjusted models. Another group of researchers who systemically reviewed the role of fruit and vegetable intakes with mental health concluded that intake of berries, citrus, and leafy greens had the best support for improving mental health, specifically depressive symptoms [31]. This conclusion is more aligned with our findings, as we found fruit intake to be inversely associated with days with depression, albeit we did not find an association between leafy green intake with days with depression or anxiety in any regression model. Lastly, a scoping review of dietary intake and anxiety reported that greater fruit and vegetable intakes were related to decreased anxiety, whereas inadequate protein intake and high fat diets were positively associated with anxiety [32]. Our findings support these associations described in the scoping review, as fruit and lean protein intake were inversely associated with anxiety, while processed meat intake, a source of saturated fat [15], was positively associated with anxiety.

This research had several strengths. First, we examined the association between dietary intake with days with depression and anxiety. There is less research regarding dietary intake and anxiety compared to depression; this is not only the conclusion of those who performed the aforementioned scoping review of dietary intake and anxiety [32], but is also reflected by the fact that there were not enough studies published on this relationship to perform a meta-analysis of the effects of dietary fiber on anxiety [9]. Secondly, in our fully adjusted models examining the association between dietary intake and depression or anxiety, age, sex, race and ethnicity, educational level attained, household income and cohabitation status were controlled for, meaning associations were not due to these potential confounders. Controlling these covariates is important for establishing genuine associations between dietary intake and mental health. When considering cohabitation status, for example, living alone has been associated with depression and anxiety [33] in addition to decreased dietary intake [34,35,36]. Another strength of this research investigating rural mental health was that it was performed in one of the most rural states in the United States, which is a unique population for this type of study. Lastly, a final strength of this work is its fairly large sample size, with close to 600 participants included in the fully adjusted models.

### Limitations

There are some limitations to this research as well. The data are cross-sectional and therefore our findings are correlational. For instance, it is unclear from these results if dietary intake affects depression and anxiety or vice versa. The sample was a convenience sample of those who responded to our recruitment methods and thus is not representative of the entire state of North Dakota. For instance, our sample was 54.5% female, whereas ND has more males than females [18]. That being said, the percentages of non-Hispanic whites in our sample in both rural and metro counties (Rural: 76.4%; Metro: 88.0%) are close to statewide estimates (Rural: 83.1%; Metro: 85.4%) [21], as is our median household income of USD 75,000–99,999 which is close to median income for ND of USD 73,240 [18]. Finally, more robust methods of assessing dietary intake and mental health could have been utilized. A three-day food diary for dietary intake assessment would have given more complete and accurate estimates of dietary intakes [37], and the use of more complete questionnaires as opposed to single items to assess both anxiety and depression, such as the Geriatric Anxiety Inventory [38] or the Center for Epidemiologic Studies Depression Scale [39], would have bolstered the veracity of our findings. However, these more robust methods would have placed greater burden on participants, potentially leading to fewer responses [40].

## 5. Conclusions

In conclusion, we found that those 50 years or older living in rural North Dakota counties reported a greater number of days where they were unable perform normal daily activities due to mental health than those living in metro counties. We also found that both days with anxiety and depression were related to days where participants reported being unable to perform normal daily activities due to mental health. While a variety of environmental factors play a role in mental health disparities between rural and metro communities [26], our findings highlight that dietary intake seems to be one of these factors that drives disparities in mental health between these areas. Those in rural counties ate more processed meat and trended toward a decreased intake of lean protein. Importantly, we found that lean protein intake was inversely associated with days with depression and anxiety, whereas processed meat intake was positively associated with anxiety. As a result, those providing dietary advice to rural middle aged and older adults should emphasize the substitution of lean proteins for processed meats.

## Figures and Tables

**Figure 1 nutrients-16-03056-f001:**
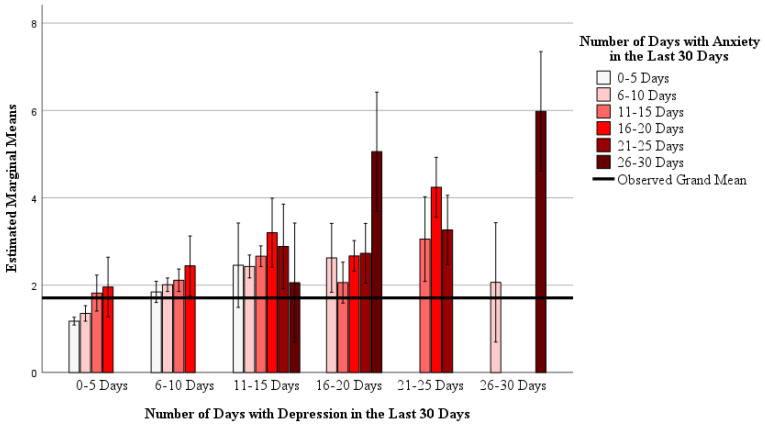
The interaction between self-reported days with depression and days with anxiety, with the number of self-reported “not good” mental health days. A mixed general linear model was used (Model: R_2_ = 0.501, F_29,559_ = 19.365, *p* < 0.001; interaction: F_13,559_ = 1.970, *p* = 0.021; main effect of depression: F_5,559_ = 9.720, *p* < 0.001; main effect of anxiety = 5.606 *p* < 0.001). Data are estimated marginal means on a 1 to 6 ordinal scale (1 = 0–5 days, 2 = 6–10 days, 3 = 11–15 days, 4 = 16–20 days, 5 = 21–25 days, and 6 = 26–30 days) controlling for age, sex, race and ethnicity, cohabitation status, education level, and household income as covariates.

**Figure 2 nutrients-16-03056-f002:**
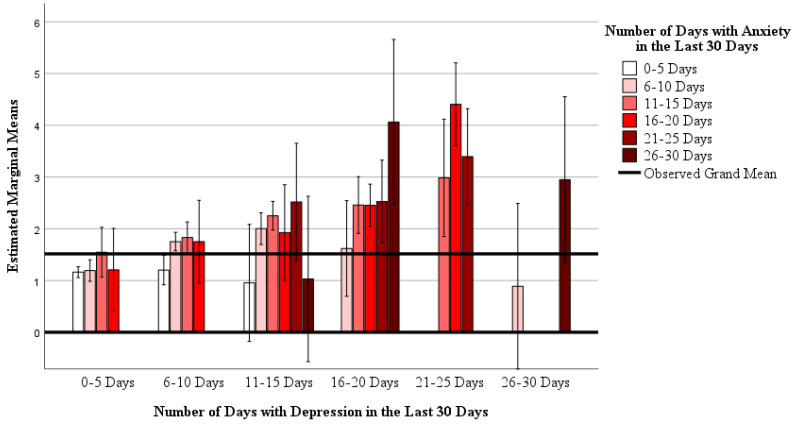
The interaction between self-reported days with depression and days with anxiety, with the number of self-reported days being unable to do normal activities due to mental health. A mixed general linear model was used (Model: R^2^ = 0.328, F_29,559_ = 9.417, *p* < 0.001; interaction: F_13,559_ = 1.849, *p* = 0.033; main effect of depression: F_5,559_ = 7.214, *p* < 0.001; main effect of anxiety = 2.304, *p* = 0.043). Data are estimated marginal means on a 1 to 6 ordinal scale (1 = 0–5 days, 2 = 6–10 days, 3 = 11–15 days, 4 = 16–20 days, 5 = 21–25 days, and 6 = 26–30 days) controlling for age, sex, race and ethnicity, cohabitation status, education level, and household income as covariates.

**Table 1 nutrients-16-03056-t001:** Demographic characteristics of the 637-person sample in rural or metro counties by frequency of response.

Age																																																																																																																																																																								
	50–55 Years																												56–61 Years																												62–64 Years																												65–70 Years																												≥71 Years																												Missing or Prefer Not to Answer																											
**Rural**	165 (32.7%)																												151 (30.0%)																												103 (20.4%)																												53 (10.5%)																												31 (6.2%)																												1 (0.2%)																											
**Metro**	32 (24.1%)																												34 (25.6%)																												18 (13.5%)																												21 (15.8%)																												28 (21.1%)																												0 (0.0%)																											
**Sex**																																																																																																																																																																								
	**Female**																																										**Male**																																										**Other**																																										**Missing or Prefer Not to Answer**																																									
**Rural**	265 (52.6%)																																										232 (46.0%)																																										6 (1.2%)																																										1 (0.2%)																																									
**Metro**	82 (61.7%)																																										50 (37.6%)																																										0 (0.0%)																																										1 (0.8%)																																									
**Race and Ethnicity**																																																																																																																																																																								
	**African American**																								**American Indian/Alaska Native**																								**Asian American/Pacific Islander**																								**Hispanic**																								**Non-Hispanic White**																								**Other**																								**Missing or Prefer Not to Answer**																							
**Rural**	48 (9.5%)																								15 (3.0%)																								21 (4.2%)																								25 (5.0%)																								385 (76.4%)																								10 (2.0%)																								0 (0.0%)																							
**Metro**	1 (0.8%)																								2 (1.5%)																								2 (1.5%)																								5 (3.8%)																								133 (88.0%)																								6 (4.5%)																								0 (0.0%)																							
**Education**																																																																																																																																																																								
	**Did Not Finish High School**																					**High School Graduate**																					**Some College**																					**Associate’s Degree**																					**Bachelor’s Degree**																					**Some Graduate School**																					**Post-Graduate Degree**																					**Missing or Prefer Not to Answer**																				
**Rural**	13 (2.6%)																					76 (15.1%)																					123 (24.4%)																					101 (20.0%)																					140 (27.8%)																					26 (5.2%)																					25 (5.0%)																					0 (0.0%)																				
**Metro**	1 (0.8%)																					0 (7.5%)																					24 (18.0%)																					21 (15.8%)																					46 (34.6%)																					7 (5.3%)																					24 (18.0%)																					0 (0.0%)																				
**Household Income**																																																																																																																																																																								
	**USD 0–24,999**																								**USD 25,000–49,999**																								**USD 50,000–74,999**																								**USD 75,000–99,999**																								**USD 100,000–149,999**																								**≥USD 150,000**																								**Missing or Prefer Not to Answer**																							
**Rural**	18 (3.6%)																								70 (13.9%)																								113 (22.4%)																								154 (30.6%)																								78 (15.5%)																								43 (8.5%)																								28 (5.6%)																							
**Metro**	6 (4.5%)																								19 (14.3%)																								27 (20.3%)																								34 (25.6%)																								25 (18.8%)																								10 (7.5%)																								12 (9.0%)																							
**Cohabitation**																																																																																																																																																																								
	**Lives Alone**																																										**Lives with a Spouse or Partner**																																										**Lives with Other Persons**																																										**Missing or Prefer Not to Answer**																																									
**Rural**	105 (20.8%)																																										384 (76.2%)																																										15 (3.0%)																																										0 (0.0%)																																									
**Metro**	35 (26.3%)																																										95 (71.4%)																																										3 (2.3%)																																										0 (0.0%)																																									

**Table 2 nutrients-16-03056-t002:** Mental health characteristics of the 637-person sample in rural or metro counties by frequency of response.

Regarding Your MENTAL Health (Stress, Depression, Emotions, etc.) in the Past 30 Days, How Many Days Would You Consider Your Mental Health “Not Good?”							
	**0–5 Days**	6–10 Days	11–15 Days	16–20 Days	21–25 Days	26–30 Days	Missing or Prefer Not to Answer
**Rural**	292 (57.9%)	111 (22.0%)	71 (14.1%)	19 (3.8%)	11 (2.2%)	0 (0.0%)	0 (0.0%)
**Metro**	78 (58.6%)	24 (18.0%)	24 (18.0%)	4 (3.8%)	1 (0.8%)	1 (0.8%)	0 (0.0%)
**Regarding Your MENTAL Health, in the Past 30 Days, How Many Days Were You Unable to Perform Your Usual, Daily Activities (Self-Care, Work, Recreation, etc.)?**							
	**0–5 Days**	**6–10 Days**	**11–15 Days**	**16–20 Days**	**21–25 Days**	**26–30 Days**	**Missing or Prefer Not to Answer**
**Rural**	345 (68.5%)	85 (16.9%)	41 (8.1%)	22 (4.4%)	8 (1.6%)	3 (0.6%)	0 (0.0%)
**Metro**	107 (80.5%)	17 (12.8%)	4 (3.0%)	3 (2.3%)	0 (0.0%)	2 (1.5%)	0 (0.0%)
**In the Past 30 Days, for How Many Days Have you Felt SAD, BLUE, or DEPRESSED?**							
	**0–5 Days**	**6–10 Days**	**11–15 Days**	**16–20 Days**	**21–25 Days**	**26–30 Days**	**Missing or Prefer Not to Answer**
**Rural**	286 (56.7%)	119 (23.6%)	61 (12.1%)	30 (6.0%)	7 (1.4%)	1 (0.2%)	0 (0.0%)
**Metro**	83 (62.4%)	27 (20.3%)	16 (12.0%)	4 (3.0%)	2 (1.5%)	1 (0.8%)	0 (0.0%)
**In the Past 30 Days, for How Many Days Have You Felt WORRIED, TENSE, or ANXIOUS?**							
	**0–5 Days**	**6–10 Days**	**11–15 Days**	**16–20 Days**	**21–25 Days**	**26–30 Days**	**Missing or Prefer Not to Answer**
**Rural**	246 (48.8%)	146 (29.0%)	76 (15.1%)	28 (5.6%)	7 (1.4%)	1 (0.2%)	0 (0.0%)
**Metro**	77 (57.9%)	30 (22.6%)	16 (12.0%)	4 (3.0%)	3 (2.3%)	3 (2.3%)	0 (0.0%)

**Table 3 nutrients-16-03056-t003:** Dietary intake characteristics of the 637-person sample in rural or metro counties by frequency of response.

How Often Do You Eat a Meal or Snack That Includes a Source of Lean Protein (Lean Meat, Poultry, Fish and Other Seafood, Eggs, Dairy, Soy and Other Beans, and Other Substitutes)?								
	Never	Rarely	Less Than Once per Week	About Once per Week	Every Two or Three Days	Once per day	More Than Once per Day	Missing or Prefer Not to Answer
**Rural**	1 (0.2%)	7 (1.4%)	5 (1.0%)	44 (8.7%)	158 (31.3%)	166 (32.9%)	123 (24.4%)	0 (0.0%)
**Metro**	1 (0.8%)	1 (0.8%)	4 (3.0%)	6 (4.5%)	41 (30.8%)	36 (27.1%)	44 (33.1%)	0 (0.0%)
	**How Often Do You Eat a Meal or Snack That Includes a Processed Meat (Bacon, Sausage, Cold Cuts, Deli Meats, etc.)?**							
	**Never**	**Rarely**	**Less Than Once per Week**	**About Once per Week**	**Every Two or Three Days**	**Once per Day**	**More Than Once per Day**	**Missing or Prefer Not to Answer**
**Rural**	4 (0.8%)	35 (6.9%)	55 (10.9%)	112 (22.2%)	139 (27.6%)	115 (22.8%)	44 (8.7%)	0 (0.0%)
**Metro**	4 (3.0%)	12 (9.0%)	25 (18.8%)	23 (17.3%)	44 (33.1%)	20 (15.0%)	5 (3.8%)	0 (0.0%)
**How Often Do You Eat a Meal or Snack That Includes a Fruit?**								
	**Never**	**Rarely**	**Less Than Once per Week**	**About Once per Week**	**Every Two or Three Days**	**Once per Day**	**More Than Once per Day**	**Missing or Prefer Not to Answer**
**Rural**	2 (0.4%)	15 (3.0%)	34 (6.7%)	36 (7.1%)	115 (22.8%)	190 (37.7%)	112 (22.2%)	0 (0.0%)
**Metro**	2 (1.5%)	5 (3.8%)	3 (2.3%)	8 (6.0%)	29 (21.8%)	46 (34.6%)	40 (30.1%)	0 (0.0%)
**How Often Do You Eat a Meal or Snack That Includes a Vegetable?**								
	**Never**	**Rarely**	**Less Than Once per Week**	**About Once per Week**	**Every Two or Three Days**	**Once per Day**	**More Than Once per Day**	**Missing or Prefer Not to Answer**
**Rural**	2 (0.4%)	7 (1.4%)	13 (2.6%)	45 (8.9%)	101 (20.0%)	175 (34.7%)	161 (31.9%)	0 (0.0%)
**Metro**	1 (0.8%)	2 (1.5%)	5 (3.8%)	3 (2.3%)	29 (21.8%)	44 (33.1%)	49 (36.8%)	0 (0.0%)
**How Often Do You Eat a Meal or Snack That Includes a Leafy, Green Vegetable (Spinach, Kale, Lettuce, etc.)?**								
	**Never**	**Rarely**	**Less Than Once per Week**	**About Once per Week**	**Every Two or Three Days**	**Once per Day**	**More Than Once per Day**	**Missing or Prefer Not to Answer**
**Rural**	3 (0.6%)	12 (2.4%)	29 (5.8%)	50 (9.9%)	149 (29.6%)	166 (32.9%)	95 (18.8%)	0 (0.0%)
**Metro**	3 (2.3%)	4 (3.0%)	6 (4.5%)	9 (6.8%)	52 (39.1%)	37 (27.8%)	2 (16.5%)	0 (0.0%)
**How Often Do You Eat a Meal or Snack That Includes Nuts and Seeds?**								
	**Never**	**Rarely**	**Less Than Once per Week**	**About Once per Week**	**Every Two Or Three Days**	**Once per Day**	**More Than Once per Day**	**Missing or Prefer Not to Answer**
**Rural**	3 (0.6%)	32 (6.3%)	51 (10.1%)	119 (23.6%)	147 (29.2%)	118 (23.4%)	34 (6.7%)	0 (0.0%)
**Metro**	3 (2.3%)	14 (10.5%)	9 (6.8%)	23 (17.3%)	34 (25.6%)	35 (26.3%)	15 (11.3%)	0 (0.0%)
**How Often Do You Eat a Meal or Snack That Includes Beans (Kidney Beans, Chickpeas, Lentils, etc.)?**								
	**Never**	**Rarely**	**Less Than Once per Week**	**About Once per Week**	**Every Two or Three Days**	**Once per Day**	**More Than Once per Day**	**Missing or Prefer Not to Answer**
**Rural**	11 (2.2%)	35 (6.9%)	79 (15.7%)	117 (23.2%)	167 (33.1%)	74 (14.7%)	21 (4.2%)	0 (0.0%)
**Metro**	4 (3.0%)	22 (16.5%)	21 (15.8%)	33 (24.8%)	35 (26.3%)	17 (12.8%)	1 (0.8%)	0 (0.0%)

**Table 4 nutrients-16-03056-t004:** Non-parametric comparisons of rural and metro counties.

Variable	Location	Median or Percentage	Interquartile Range (Min, Max)	Χ^2^ or U ^a^ (*p*)
Age (Ordinal)	Rural	2.00	1.00–3.00 (1, 5)	7.099 (*p* = 0.008)
	Metro	3.00	2.00–3.00 (1, 5)	
Sex (Categorical)	Rural	46.0% Male	-	−1.806 (*p* = 0.071)
	Metro	37.6% Male	-	
Race and Ethnicity (Categorical)	Rural	76.4% non-Hispanic white	-	3.458 (*p* < 0.001)
	Metro	88.0% non-Hispanic white	-	
Education (Ordinal)	Rural	4.00	3.00–5.00 (1, 7)	16.457 (*p* < 0.001)
	Metro	5.00	3.00–5.00 (1, 7)	
Household Income (Ordinal)	Rural	4.00	3.00–5.00 (1, 6)	0.446 (*p* = 0.504)
	Metro	4.00	3.00–5.00 (1, 6)	
Cohabitation (Categorical)	Rural	20.8% Live alone	-	1.357 (*p* = 0.175)
	Metro	26.3% Live alone	-	
Number of “Not Good” Mental Health Days in the Last 30 Days (Ordinal)	Rural	1.00	1.00–2.00 (1, 6)	0.002 (*p* = 0.961)
	Metro	1.00	1.00–2.00 (1, 6)	
Number of Days Unable to Perform Activities due to Poor Mental Health (Ordinal)	Rural	1.00	1.00–2.00 (1, 6)	6.781 (*p* = 0.009)
	Metro	1.00	1.00–1.00 (1, 6)	
Number of Days with Depressive Symptoms in the Last 30 Days (Ordinal)	Rural	1.00	1.00–2.00 (1, 6)	1.161 (*p* = 0.281)
	Metro	1.00	1.00–2.00 (1, 6)	
Number of Days with Anxiety Symptoms in the Last 30 Days (Ordinal)	Rural	2.00	1.00–2.00 (1, 6)	3.121 (*p* = 0.077)
	Metro	1.00	1.00–2.00 (1, 6)	
Lean Protein Intake (Ordinal)	Rural	6.00	5.00–6.00 (1, 7)	3.660 (*p* = 0.056)
	Metro	6.00	5.00–7.00 (1, 7)	
Processed Meat Intake (Ordinal)	Rural	5.00	4.00–6.00 (1, 7)	7.719 (*p* = 0.005)
	Metro	5.00	3.00–5.00 (1, 7)	
Fruit Intake (Ordinal)	Rural	6.00	5.00–6.00 (1, 7)	3.153 (*p* = 0.076)
	Metro	6.00	5.00–7.00 (1, 7)	
Vegetable Intake (Ordinal)	Rural	6.00	5.00–7.00 (1, 7)	0.931 (*p* = 0.335)
	Metro	6.00	5.00–7.00 (1, 7)	
Leafy Green Intake (Ordinal)	Rural	6.00	5.00–6.00 (1, 7)	0.236 (*p* = 0.627)
	Metro	5.00	5.00–6.00 (1, 7)	
Nuts and Seeds Intake (Ordinal)	Rural	5.00	4.00–6.00 (1, 7)	2.354 (*p* = 0.125)
	Metro	5.00	4.00–6.00 (1, 7)	
Beans Intake (Ordinal)	Rural	5.00	4.00–5.00 (1, 7)	5.723 (*p* = 0.017)
	Metro	4.00	3.00–5.00 (1, 7)	

^a^ Ordinal variables were compared using Mood’s test of medians, whereas categorical variables were evaluated using the Mann–Whitney U test. Χ^2^ values are given for Mood’s test of medians, and the standardized test statistic is given for the Mann–Whitney U test.

**Table 5 nutrients-16-03056-t005:** The Association Between Dietary Intake and Self-Reported Days with Depressive Symptoms.

Dietary Intake Variable	Univariate (*n* = 637)		Partially Adjusted Model Including Demographic Covariates (*n* = 589) ^a^		Fully Adjusted Model Including Dietary Covariates (*n* = 589) ^b^	
	β ± S.E. ^c^	*p*	β ± S.E.	*p*	β ± S.E.	*p*
**Lean protein intake**	−0.189 ± 0.036	<0.001	−0.146 ± 0.037	<0.001	−0.117 ± 0.040	0.004
**Processed meat intake**	0.043 ± 0.028	0.133	−0.010 ± 0.030	0.741	0.016 ± 0.032	0.614
**Fruit intake**	−0.139 ± 0.030	<0.001	−0.093 ± 0.032	0.004	−0.096 ± 0.038	0.011
**Vegetable intake**	−0.120 ± 0.036	<0.001	−0.055 ± 0.035	0.115	−0.027 ± 0.044	0.536
**Leafy green intake**	0.021 ± 0.031	=0.512	0.039 ± 0.033	0.232	0.078 ± 0.041	0.059
**Nut and seed intake**	−0.021 ± 0.029	=0.512	−0.022 ± 0.030	0.461	−0.003 ± 0.035	0.925
**Bean intake**	0.058 ± 0.029	<0.001	0.003 ± 0.031	0.920	0.014 ± 0.037	0.712

^a^ The partially adjusted model included age, sex, race and ethnicity, cohabitation status, education level, and household income as covariates. ^b^ The fully adjusted model included all covariates from the partially adjusted model and all dietary intake variables. ^c^ S.E. = Standard error.

**Table 6 nutrients-16-03056-t006:** The Association Between Dietary Intake and Self-Reported Days with Anxiety Symptoms.

Dietary Intake Variable	Univariate (*n* = 637)		Partially Adjusted Model Including Demographic Covariates (*n* = 589) ^a^		Fully Adjusted Model Including Dietary Covariates (*n* = 589) ^b^	
	β ± S.E. ^c^	*p*	β ± S.E.	*p*	β ± S.E.	*p*
**Lean protein intake**	−0.145 ± 0.037	<0.001	−0.108 ± 0.037	0.004	−0.086 ± 0.040	0.031
**Processed meat intake**	0.086 ± 0.029	0.003	0.044 ± 0.30	0.146	0.089 ± 0.040	0.005
**Fruit intake**	−0.157 ± 0.031	<0.001	−0.105 ± 0.032	0.001	−0.095 ± 0.038	0.012
**Vegetable intake**	−0.138 ± 0.034	<0.001	−0.089 ± 0.035	0.011	−0.051 ± 0.044	0.242
**Leafy green intake**	−0.012 ± 0.032	0.712	−0.006 ± 0.033	0.853	0.068 ± 0.041	0.103
**Nut and seed intake**	−0.028 ± 0.30	0.356	−0.022 ± 0.30	0.452	0.032 ± 0.035	0.929
**Bean intake**	−0.003 ± 0.030	0.916	−0.070 ± 0.031	0.025	−0.092 ± 0.037	0.013

^a^ The partially adjusted model included age, sex, race and ethnicity, cohabitation status, education level, and household income as covariates. ^b^ The fully adjusted model included all covariates from the partially adjusted model and all dietary intake variables. ^c^ S.E. = Standard error.

## Data Availability

Due to data sharing policies related to IRB permission requirements for studies involving adults aged 50 and older, data cannot be posted as a Appendix A at this time. Permissions from university IRB have not been obtained for data sharing. Instead, readers should contact the corresponding author to access the data or for further inquiries.

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
