# Peer review of "Intakes of Lean Proteins and Processed Meats and Differences in Mental Health between Rural and Metro Adults 50 Years and Older"

_nutrients, 2024, doi:10.3390/nu16183056_

Round 1
Reviewer 1 Report
Comments and Suggestions for Authors
General Comments
The research topic is interesting and looks into a dimension that might have a greater impact on public health, particularly focusing on a segment of the US population that is not being addressed in research. The study approach and methodology are great.
However, many aspects need attention if this is to be published in Nutrients.
The title might be adjusted to avoid confusion - e.g. the meanings of "lean protein" and "processed meat" are confusing. "Lean meat is fat read skeletal muscle", while processed meat is any meat including lean meat preserved or cured in some way, that doesn't necessarily mean 'bad'. There are repeated words such as "Differential" vs "Differences";
It is not clear in the write-up what "Lean proteins" and "processed meat" are and this might necessitate the inclusion of a glossary of terms with a complete list of examples (including foods) in each group/category. This might also be needed for what mental health is and what symptoms are used to indicate depression and anxiety.
Multiple instances show a lack of focus - where information being discussed is way off-topic (e.g., dietary fiber (lines 35-28), fat (54-57), ultra-processed foods (lines 268-269), fruits and vegetables (lines 278-279), etc. in a research topic discussing protein intake.
Tables 1 through 3 need to be explained and summarized under section 3.1.
The first paragraph of the discussion is shallow and sounds more like a conclusion and recommendation section - more depth is needed on what the implications of the results are.
Line 268: Q2. What does "ultra-processed" food mean? What example was reported in the current survey how is that connected to the topic? What do high-glycemic starches and sugars, have to do with "processed meat"?
The last two paragraphs of the discussion section must be organized in a separate topic "Opportunities limitations".
Specific Comments
Line 5: Please exclude the qualifications and other extra descriptors from the Authors' names.
Lines 15-16: It would make better sense if "dietary intake" comes before "mental health"; Also, change "between" to "among" or rephrase the statement to make the "between refer to the two residence areas.
Lines 20-23: it is not clear if the authors are suggesting (based on data) that dietary intake resulted in mental health or otherwise. Rephrasing might help to ensure better clarity.
Line 24; delete the first "intake"
Line 31: change the first disease to "health issues"
Lines 35-38: The statement is too wordy and hard to follow with so many words repeating in the same statement ("dietary" = 3 times; "intake" = 2 times; The statement begins with plural citation "Others reported that ..." but only a single article is cited at the end - suggesting at least two relevant citations.
Lines 47-49: The statement is confusing - either add "only" before "about half ..." or mention the percentage of adults not meet the intake recommendation for both protein and fiber.
Lines 50-54: The statement is mistakenly broken into two paragraphs; The statement is also too long and structured in confusing language - needs attention! Also, insert "in" after "living" in line 52.
Lines 78-80: The Statement is wordy with repeated expressions in the same statement (e.g. "dietary intake" = 2 times; "to examine" = 2 times).
Line 100: change the second "the survey" to "it" to avoid repeating similar words in the same statement.
Line 134: "Questionnaire" is repeated in the same statement.
Lines 177-178: Check the accuracy of the term "graphical abstract" - not commonly made with a model, Plus, I didn't see any graphical abstract included in the draft.
Lines 203 and 230 have similar section numbering (3.3.) and should be corrected.
Lines 260-263: Q1. How is the recommended increase of grocery stores practically possible when only sparsely and distantly populated residents live and when groceries have limited shelf life?
Comments on the Quality of English Language
Manuscript might benefit from language editing
Author Response
Please see the attached file for our responses to your feedback.

Reviewer 2 Report
Comments and Suggestions for Authors
Thank you for submitting the manuscript "Differential Intakes of Lean Proteins and Processed Meats Drive Differences in Mental Health Between Rural and Metro Adults 50 Years and Older" to Nutrients. Although the manuscript is interesting for assessing food consumption and (self-reported) mental state of two types of population, it fails to use only the questionnaire as a data collection tool. Obviously, the number of samples collected was huge and this justifies the population cut, but the authors could compare the results obtained by self-report with government data, for example. Another issue is that the results item needs to present the results of facts and not just present tables and figures and let the reader read them.
- It is interesting to add the number of people who were used in the study right away in the abstract.
- Line #35: consider putting a comma before respectively.
- Line #45: it is important to emphasize that they normally consume as little food as they should (considering nutrient needs). It is also important to consider that many older adults feel depressed because they have abandoned their families. - Consider dividing the material and methods item into subitems, such as "study protocol" and "statistical analysis" for better clarity.
- Line #182: it would be interesting to include the date of collection of the questionnaire (start and end). It is important to highlight that the results of Tables 1, 2 and 3 were not presented. What was done was the presentation of the tables and therefore it needs to be done. This needs to be checked for the entire results item.
- Line #195 and others: check the use of punctuation throughout the text as it is missing.
- Lines #200-201: reword to improve clarity.
Comments on the Quality of English LanguageMinor editing of English language required.
Author Response

(The authors gave the same response as above.)

Round 2
Reviewer 1 Report
Comments and Suggestions for Authors
Comments and Concerns are addressed and others are explained well. The manuscript is way better than the original submission.
Comments on the Quality of English LanguageIt is better, but the article might benefit from a review by a native English speaker.
Reviewer 2 Report
Comments and Suggestions for Authors
This review thanks the authors who made a great effort to review the entire manuscript and make the suggested corrections. Therefore, my suggestion is to accept the manuscript for publication.